# Autocorrelation structure at rest predicts value correlates of single neurons during reward-guided choice

Sean E Cavanagh[1], Joni D Wallis[2,3], Steven W Kennerley[1,2,3*†], Laurence T Hunt[1,4*†]

[1]Sobell Department of Motor Neuroscience, University College London, London, United Kingdom; [2]Department of Psychology, University of California, Berkeley, Berkeley, United States; [3]Helen Wills Neuroscience Institute, University of California, Berkeley, Berkeley, United States; [4]Wellcome Trust Centre for Neuroimaging, University College London, London, United Kingdom

**Abstract** Correlates of value are routinely observed in the prefrontal cortex (PFC) during reward-guided decision making. In previous work (Hunt et al., 2015), we argued that PFC correlates of chosen value are a consequence of varying rates of a dynamical evidence accumulation process. Yet within PFC, there is substantial variability in chosen value correlates across individual neurons. Here we show that this variability is explained by neurons having different temporal receptive fields of integration, indexed by examining neuronal spike rate autocorrelation structure whilst at rest. We find that neurons with protracted resting temporal receptive fields exhibit stronger chosen value correlates during choice. Within orbitofrontal cortex, these neurons also sustain coding of chosen value from choice through the delivery of reward, providing a potential neural mechanism for maintaining predictions and updating stored values during learning. These findings reveal that within PFC, variability in temporal specialisation across neurons predicts involvement in specific decision-making computations.

*For correspondence:
s.kennerley@ucl.ac.uk (SWK);
laurence.hunt@ucl.ac.uk (LTH)

†These authors contributed equally to this work

Competing interests: The authors declare that no competing interests exist.

## Introduction

Theoretical models of decision making emphasise the importance of evidence accumulation across time until a categorical choice is reached (*Bogacz et al., 2006*; *Gold and Shadlen, 2007*). One widely studied class of evidence accumulation models are cortical attractor networks, originally derived from studies of working memory (*Amit and Brunel, 1997*; *Wang, 1999*, *2002*). These rely upon strong recurrent connections between similarly tuned neurons to integrate evidence across time, and exhibit temporally extended persistent activity that stores the outcome of the decision process in memory (*Wang, 2002*; *Wong and Wang, 2006*). In value-guided decision making tasks, attractor network models predict the emergence of correlates of chosen value during choice (*Hunt et al., 2012*; *Rustichini and Padoa-Schioppa, 2015*). These value correlates result from varying speeds of decision formation across different trials, an issue we explored closely in our previous paper (*Hunt et al., 2015*). However, in contrast to the relative homogeneity of chosen value correlates within such models, it is known that decision correlates are highly *heterogeneous* across different cells within a given region (*Kennerley et al., 2009*; *Wallis and Kennerley, 2010*; *Meister et al., 2013*). The source and functional significance of this neuronal heterogeneity remains unclear.

Neurons also exhibit heterogeneity in their *temporal receptive fields of integration* (*Chen et al., 2015*). The temporal receptive field of a neuron can be established by examining its spike-count autocorrelation function (ACF) at rest (*Ogawa and Komatsu, 2010*). A slowly decaying ACF whilst

at rest reflects temporal stability in firing, suggesting that the neuron integrates information across long periods of time; by contrast, a fast-decaying ACF reflects temporal variability in firing. Recently, this approach was used to demonstrate a hierarchy of temporal receptive fields across areas of cortex (*Murray et al., 2014*), with populations of neurons in lower and higher cortical areas exhibiting brief and extended temporal receptive fields, respectively. Those areas with temporally extended receptive fields thus appear intrinsically adapted to cognitive tasks involving extended integration of information across time, such as working memory and decision making (*Mazurek et al., 2003*; *Gold and Shadlen, 2007*; *Wang, 2012*; *Chaudhuri et al., 2015*; *Chen et al., 2015*). Yet in addition to the heterogeneity of temporal fields *across* regions, similar heterogeneity is also evident *within* cortical areas (*Ogawa and Komatsu, 2010*; *Nishida et al., 2014*). It remains unknown whether this intra-regional heterogeneity in temporal specialisation might predict the computations served by different neurons in decision-making tasks.

In our previous study of reward-guided decision making (*Hunt et al., 2015*), we provided evidence that correlates of chosen value may emerge as a consequence of varying rates of evidence accumulation. A corollary of this idea is that neurons functionally specialised to perform temporally extended computations (such as evidence accumulation) might exhibit stronger chosen value correlates during choice. We hypothesised that this would be indexed by measuring individual neurons' temporal receptive fields whilst at rest. We also hypothesised that this functional specialisation might support other temporally extended computations during reward-guided choice, such as the maintenance of value coding until reward delivery. This could be one component of a mechanism for credit assignment in learning, which is known to rely upon PFC and in particular orbitofrontal cortex (*Walton et al., 2010*; *Takahashi et al., 2011*; *Chau et al., 2015*; *Jocham et al., 2016*), with the other component being a representation of the chosen stimulus identity, which is also encoded by OFC neurons (*Raghuraman and Padoa-Schioppa, 2014*; *Lopatina et al., 2015*). We therefore sought to link variability in spike-rate autocorrelation at rest with the variability of neuronal responses during reward-guided choices.

## Results

We re-examined the neural correlates of chosen value during choice within rhesus macaque prefrontal cortex (PFC) (*Hosokawa et al., 2013*; *Hunt et al., 2015*), and extended our analysis to the time of reward delivery (*Figure 1*, *Figure 1—figure supplement 1*). During choice, chosen value correlates were remarkably similar across all three PFC brain regions (dorsolateral prefrontal cortex (DLPFC), orbitofrontal cortex (OFC) and anterior cingulate cortex (ACC)) at the population level (*Figure 1A*). However, this was not the case at the time of outcome, where the chosen value correlates predominated in OFC (*Figure 1B*). This value signal at outcome contained information about both the chosen benefit and chosen cost (*Figure 1—figure supplement 2*). As well as variability in value correlates across time, there was a large degree of variability at the level of single neurons constituting the population averages, both at choice and outcome (*Figure 1C–D*). Within each region there were some neurons with strong chosen value correlates, but other neurons with weak or non-selective responses to chosen value.

We hypothesised that this variability might be accounted for by intrinsic firing properties of the neurons at rest, reflecting different neurons' temporal specialisation. We characterised resting properties of neuronal firing by examining their spike rate autocorrelation during pre-trial fixation. The decay of the autocorrelation function (ACF) provides a metric of each neuron's temporal stability in firing rate. Careful inspection of ACFs at the level of single neurons demonstrated marked heterogeneity of ACFs across individual neurons (*Figure 2*), complementing previous descriptions that have examined average population responses (*Murray et al., 2014*; *Chaudhuri et al., 2015*). We fitted an exponential decay function (*Murray et al., 2014*) to all neurons that could be described by such an equation, yielding a single decay time constant, $\tau$, for each neuron (446 of 857 neurons, see *Figure 2—figure supplement 1* and Materials and methods). We found a large degree of heterogeneity in time constants across neurons, both within and between cortical areas (*Figure 2C*). Time constants were larger in the DLPFC and ACC population (Kruskal-Wallis test, p=0.0007), but most variable within OFC and ACC populations (Bartlett's Statistic = 11.913, p=0.0026). Averaging across the ACFs of individual neurons prior to fitting the exponential equation yielded similar qualitative

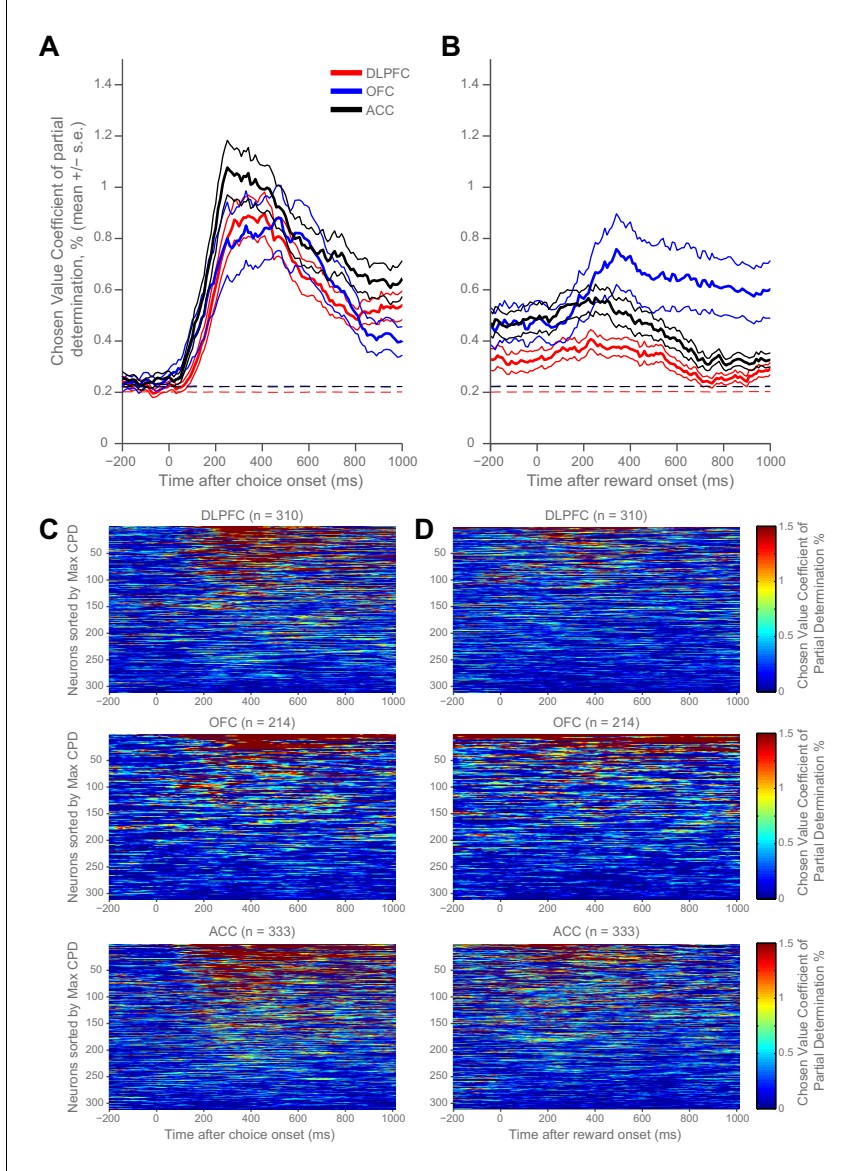

**Figure 1.** Homogeneity and heterogeneity of chosen value correlates. (**A**) At decision time, chosen value correlates appeared *homogenous* across regions in their expression. The coefficient of partial determination (CPD) for chosen value averaged across populations of DLPFC (n = 310), OFC (n = 214) and ACC (n = 333) neurons (lines denote mean ± SE for each region). CPD was calculated by regressing chosen value onto firing rate during the choice period of a cost-benefit decision making task (see Materials and methods). Chosen value correlates were not significantly different between any brain region (permutation tests; DLPFC v OFC, no cluster survived thresholding, DLPFC v ACC, p=0.2706, OFC v ACC, no cluster survived thresholding; see Materials and methods). Dashed lines mark the null hypothesis level for CPD in each cortical area (see Materials and methods). (**B**) Population averages when chosen value was regressed onto firing rate during reward delivery. OFC showed stronger chosen value correlates following reward onset than ACC and DLPFC (permutation tests; OFC v DLPFC, p=0.0010, OFC v ACC, p=0.0028; see Materials and methods). (**C** and **D**) Within each region, chosen value correlates were *heterogeneous* across neurons. Chosen value correlates of the individual neurons contributing to the population averages in **A** and **B** respectively. Within each matrix: each row is a neuron (sorted by maximum CPD within the corresponding epoch and area), each column is a 10 ms time bin. Hence, neurons are sorted in a different order in **C** and **D**. Chosen value coding at reward delivery was weaker than at choice. *Figure 1—figure supplement 1* shows the fraction of neurons with reliable coding of chosen value at choice and at the outcome. *Figure 1—figure supplement 2* shows that OFC codes chosen value, as opposed to chosen benefit alone, at the time of reward delivery.

*Figure 1 continued on next page*

*Figure 1 continued*

The following figure supplements are available for figure 1:

**Figure supplement 1.** Fraction of neurons with reliable coding of chosen value at choice (**A**) and at outcome (**B**).

**Figure supplement 2.** Orbitofrontal cortex codes chosen value, as opposed to chosen benefit alone, at the time of reward delivery.

results to the population averages reported in *Murray et al. (2014)* (*Figure 2—figure supplements 2* and *3*).

Our main question pertained to whether the observed variability in single-cell resting activity within PFC may determine different functional computations during a cost-benefit decision making task (*Hosokawa et al., 2013*; *Hunt et al., 2015*). We first sought to visually identify a potential relationship with chosen value by sorting the matrices in *Figure 1C* by time constant. To maximise our sensitivity, and because of the similarity in chosen value correlates across PFC brain regions at choice (*Figure 1A*), we collapsed this analysis across all three PFC regions (n = 446 neurons). We found that more neurons with high chosen value coefficient of partial determination (CPD) were more apparent at the bottom of the sorted matrix than at the top (*Figure 3A*), implying a relationship between chosen value coding and resting τ. To test this relationship statistically, neurons were subdivided into high and low time constant populations using a median split (*Figure 3B*). The population with a higher τ (more stable activity at rest) had more variance explained by chosen value during choice (permutation test (see Materials and methods), p=0.0298). We further demonstrated this relationship by performing a rank correlation between each neuron's coefficient of partial determination (CPD) at the time of the maximum population-average CPD with its time constant (Correlation Coefficient = 0.148, p=0.0018; 95% CI [0.0556, 0.2373], *Figure 3—figure supplement 1*). This relationship was also present when controlling for the baseline firing rate and brain area using multiple regression (see Materials and methods, β = 0.3315, p=0.0254; 95% CI [0.0878 0.5751]).

We then repeated the analysis in *Figure 3B* across all three regions. We found that the relationship between high τ and chosen coding was particularly prominent in OFC and ACC, but observed no significant difference in the chosen value coding between populations with high/low τ in DLPFC (*Figure 4*). If the chosen value correlates were purely related to the dynamics of choice processes, we might expect them to return to baseline levels after the choice had been executed. Although this was largely the case, a degree of chosen value coding persisted until reward outcome, particularly within OFC (*Figure 1B*). Within OFC, we found that persistent coding of chosen value from choice to outcome was more evident within the high τ neuronal population than within the low τ population, particularly during the experience of reward delivery (*Figure 4*; permutation test at time of outcome (see Materials and methods), p=0.0082). Such sustained coding of chosen value from choice through outcome was not present in ACC and DLPFC. This implies a unique neuronal signature within OFC which could contribute to the linking of choices to outcomes, a process critical for learning.

Given the above result, we sought to address whether the same OFC neurons were signalling chosen value at the time of both choice and outcome. We performed a cross-temporal pattern analysis on data from the OFC (*Kennerley et al., 2011*; *Stokes et al., 2013*). This involves cross-correlating the chosen value regression coefficients of the entire neuronal population at all of the different time bins. If the same neurons encode chosen value at timepoints *t* and *t+δt*, one would expect a high correlation between these two timepoints; conversely, coding of chosen value by different neural ensembles would yield a far smaller, or zero, correlation. By examining the matrix of correlation coefficients at all possible timelags, different types of population neural coding can be revealed (such as transient, reactivation, or sustained coding; see *Figure 5A*). To avoid this analysis being confounded by noise correlations, we performed a 'split half' cross correlation analysis, calculating the regression coefficients for chosen value separately for odd and even trials.

During the choice epoch, there was unsurprisingly evidence for on-diagonal coding (top left quadrant of *Figure 5B*). The OFC neuronal population code was also persistent across time during this epoch (warm *off*-diagonal elements in top left quadrant of *Figure 5B*), and even more so during

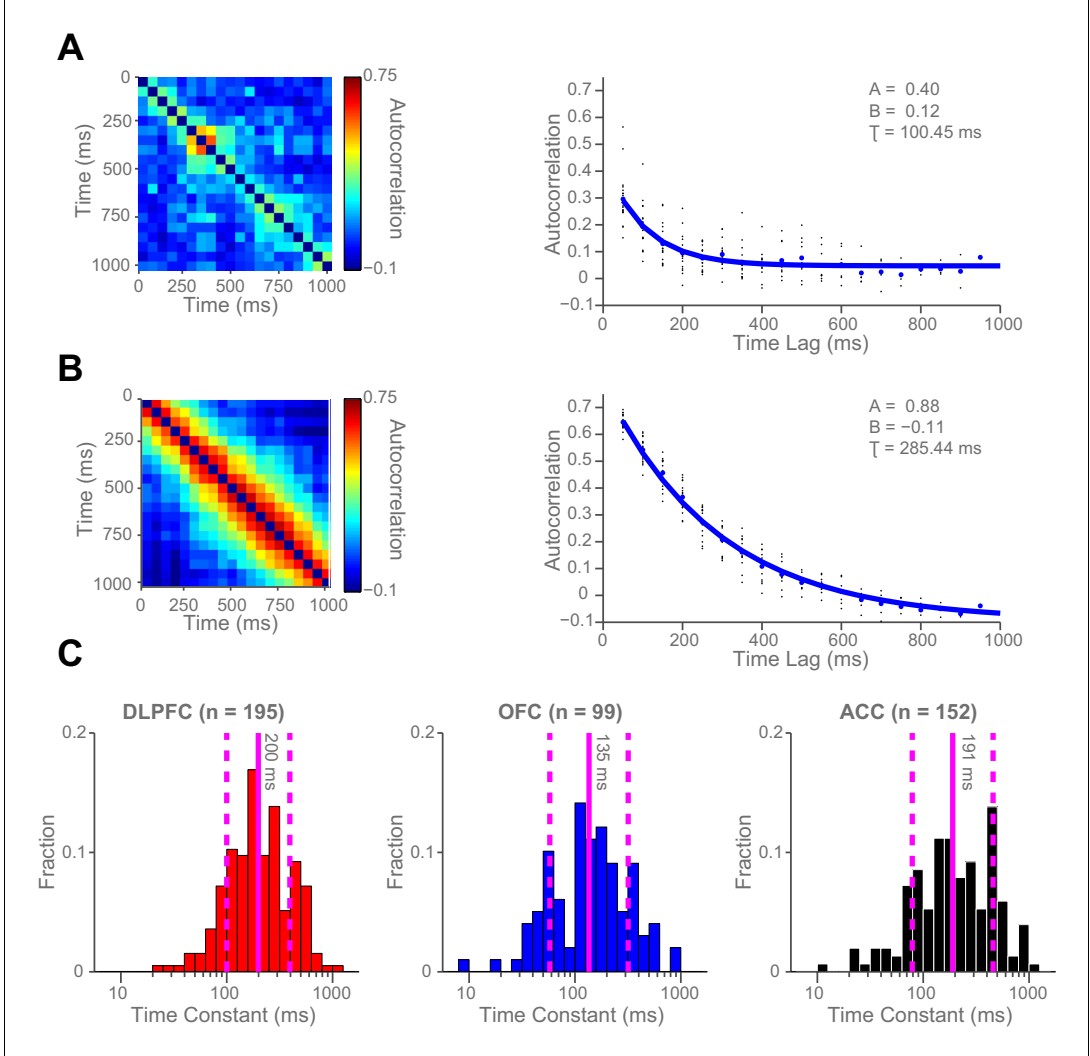

**Figure 2.** Single neurons show variability in resting autocorrelation structure. (**A**) Autocorrelation matrix and structure of an example low time constant OFC single neuron. (**B**) Autocorrelation matrix and structure of an example high time constant single OFC neuron. This neuron has a stable autocorrelation maintained across time. Fitting of time constants was only performed on cells that showed an exponentially decaying autocorrelation. See **Figure 2—figure supplement 1** for single neuron examples of excluded cells. (**C**) Histograms of the time constants within the three PFC brain regions. Time constants are highly variable across neurons; with the greatest heterogeneity present within OFC and ACC populations. Solid and dashed vertical lines represent mean(Log(τ)) and mean(Log(τ)) ± SD(Log(τ)) respectively. See **Figure 2—figure supplement 2** for autocorrelation structure at the population level. **Figure 2—figure supplement 3** for population autocorrelation when trials are filtered for fluctuations in firing rate. **Figure 2—figure supplement 4** shows the population autocorrelation across trial time.

The following figure supplements are available for figure 2:

**Figure supplement 1.** Autocorrelation of example single neurons that were excluded from all subsequent analyses.

**Figure supplement 2.** Autocorrelation structure of the DLPFC, OFC and ACC populations.

**Figure supplement 3.** The autocorrelation structure of the DLPFC, OFC and ACC population when trials were filtered for drifts in resting firing rate.

**Figure supplement 4.** Population Autocorrelation across trial time.

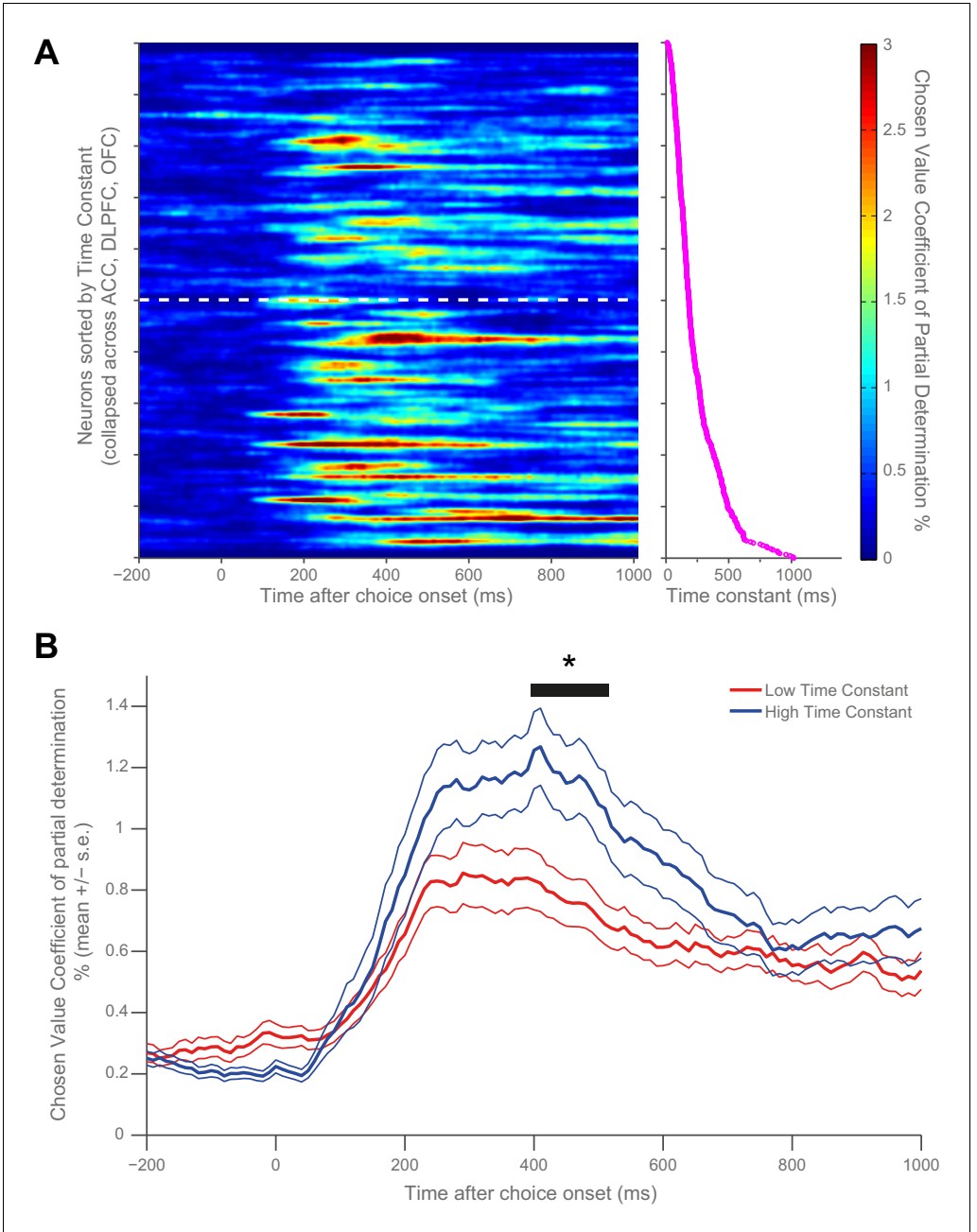

**Figure 3.** Resting time constant predicts chosen value correlates during decision phase. (**A**) Strong chosen value correlates were more prevalent in neurons with higher time constants. Coefficient of partial determination (CPD) for chosen value across time for each PFC neuron (n = 446) was stacked into a matrix. The rows of the matrix (i.e. each individual neuron) were sorted by increasing time constant, and then convolved with a Gaussian function (see Materials and methods). The white dashed line indicates a median split by time constant; high time constant neurons are beneath the line, low time constant neurons are above. The graph to the right of this matrix shows the individual decay time constant for each neuron (row) in the matrix. (**B**) When all neurons are subdivided by a median split of time constant, those with a higher time constant exhibit stronger chosen value correlates. Black trace indicates a significant cluster of bins, corrected for multiple comparisons across time (see Materials and methods, p=0.0298). CPD (mean ± SE) for chosen value was calculated by multiple linear regression analysis (see Materials and methods). *Figure 3—figure supplement 1* shows a rank correlation of resting time constant with chosen value coding across time.

The following figure supplement is available for figure 3:

*Figure 3 continued on next page*

*Figure 3 continued*

**Figure supplement 1.** Rank correlation between resting time constant and chosen value correlates during the decision phase.

the outcome epoch (warm off-diagonal elements in bottom right quadrant of *Figure 5B*). This sustained activity reflects the notion that dynamical decision processes within the OFC population may take place over several hundreds of milliseconds. Crucially, however, there was also evidence for sustained coding: the *same* neuronal population in OFC at choice encoded chosen value from at least 1000 ms before outcome through to 1000 ms after outcome (warm colours in *Figure 5B*, grey and

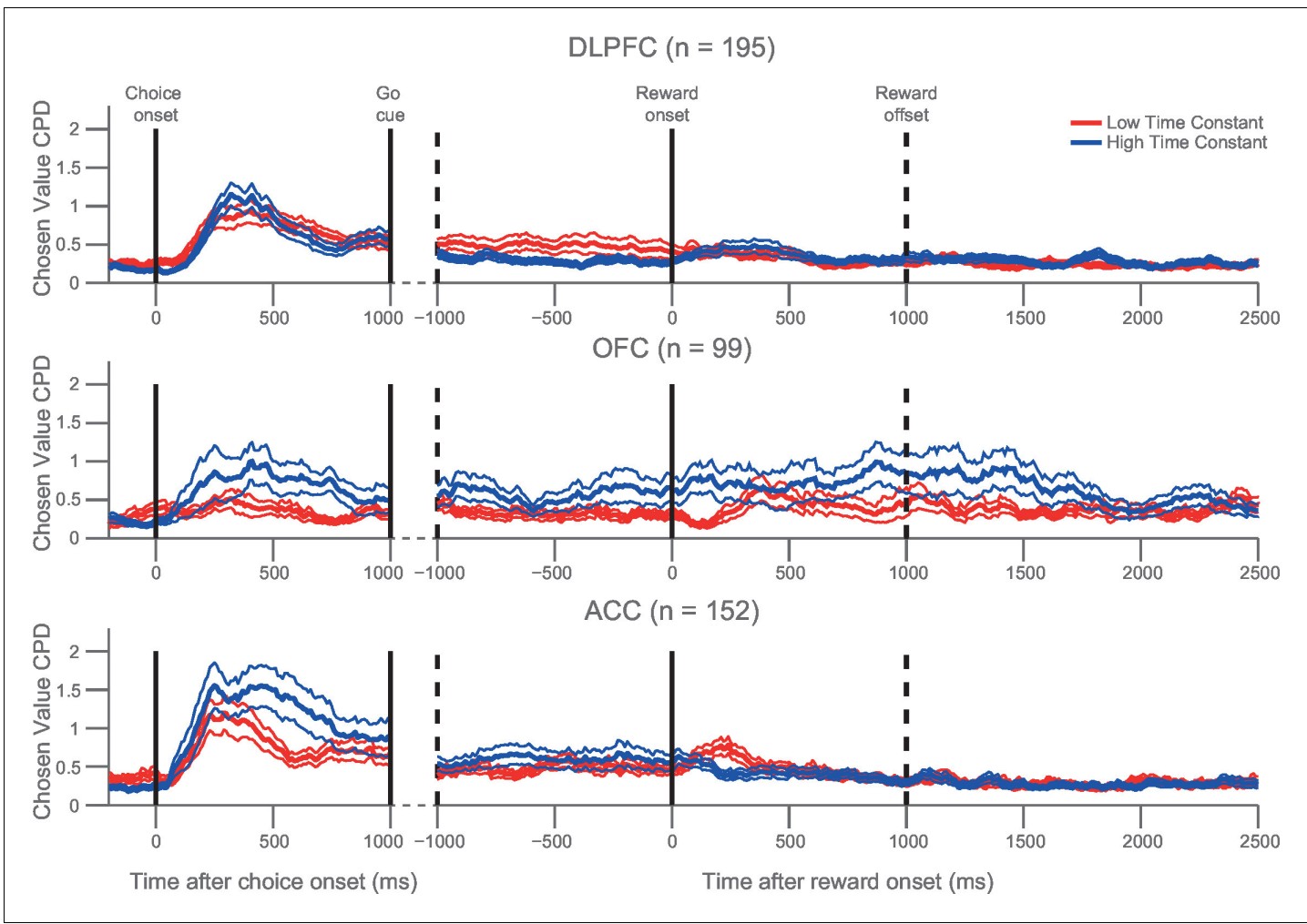

**Figure 4.** Orbitofrontal neurons with higher resting time constant maintain a representation of chosen value from choice through the experience of reward delivery. As in *Figure 3B*, a median split of neurons by their resting time constant was performed within each PFC area. The coefficient of partial determination (CPD) for chosen value in high time constant (blue) and low time constant (red) neurons is plotted timelocked to both choice and reward onset. Chosen value explained more of the variance in neuronal firing in the OFC neurons with a higher time constant both at choice (p=0.0066) and shortly after reward delivery (p=0.0082). Chosen value is therefore maintained across the trial within OFC, but returns to baseline before the next trial begins. CPD (mean ± SE) for chosen value was calculated by multiple linear regression analysis (see Materials and methods). *Figure 4—figure supplement 1* shows a rank correlation of resting time constant with chosen value coding during the decision phase and reward delivery.
The following figure supplement is available for figure 4:

**Figure supplement 1.** Rank correlation between resting time constant and chosen value correlates during choice and following reward.

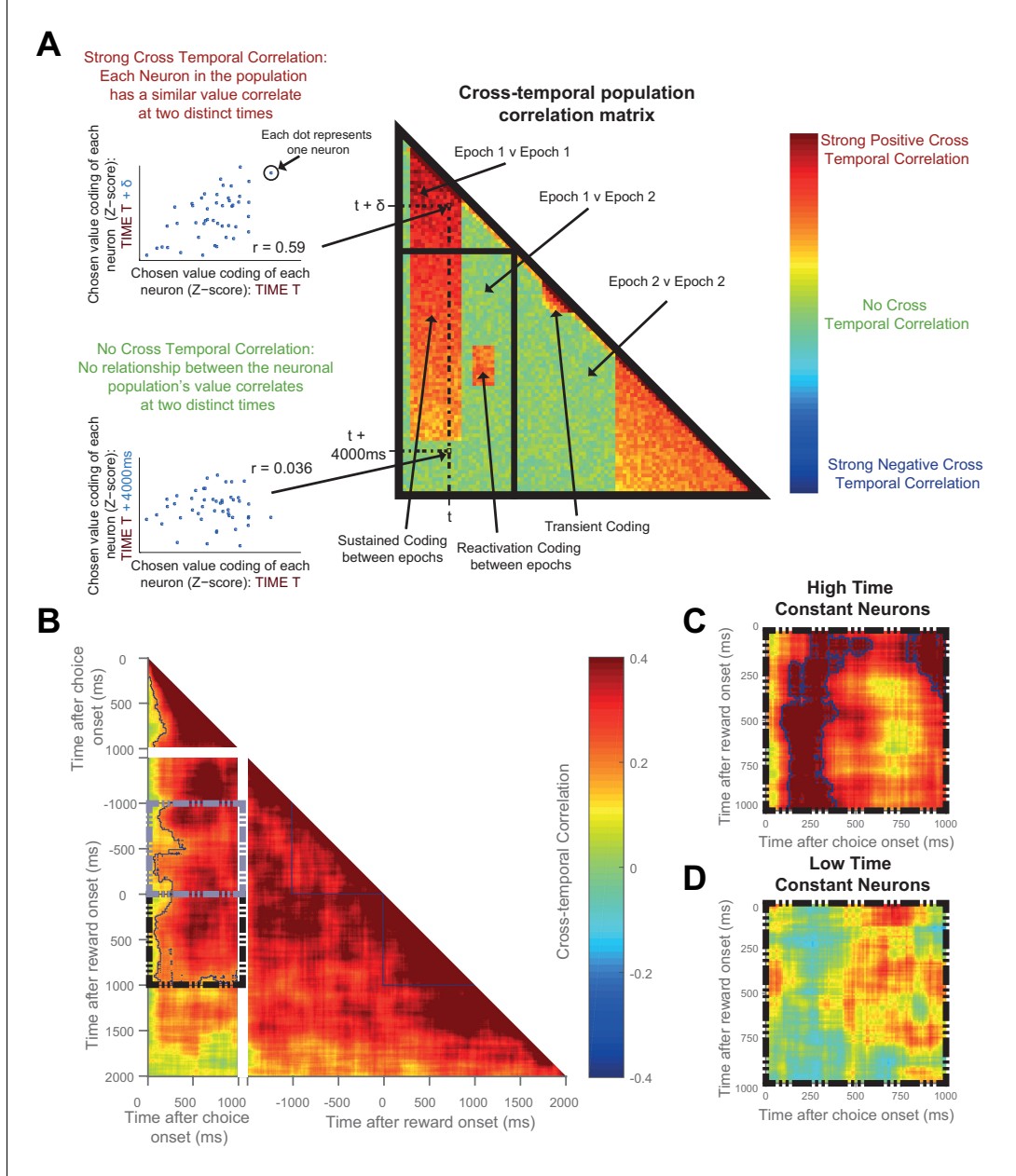

**Figure 5.** The same OFC neurons correlate strongly with chosen value at both choice and during reward delivery, but only those with high time constants. (**A**) Schematic representing the cross-temporal pattern analysis. Each pixel represents a correlation coefficient between two population vectors. Entries into the vectors contain e*ach neuron's chosen value regression coefficient* at Time T and at Time T + δt. If the chosen value correlates are consistent across the neuronal population at the two distinct time points, there will be a strong cross-temporal correlation (red colour). At two points close in time, chosen value correlates of each neuron will inevitably be similar. If these correlates are consistent for only a short period of time, there will be a *transient population code*; whereas if each neuron's chosen value correlate is consistent for a prolonged period, there will be a *sustained population code*. If each neuron within a population correlates with chosen value at two separate points of a trial (e.g. choice and outcome), in the absence of sustained coding bridging the two, there is a *reactivation population code*. (**B**) Cross-temporal pattern analysis of OFC neurons (n = 214). There is clear evidence for sustained coding of the chosen value at choice (top left), as well as before and throughout outcome (bottom right), reflected by strong correlations extending off the diagonal of the plot. Blue lines indicate a significant area of cross-correlation (p<0.05, see Materials and methods). There is also sustained coding of the chosen value signal from choice through outcome, shown by a strong cross-temporal correlation both prior (grey dashed box) and during reward (black dashed box). Within the dashed areas, blue lines indicate a significant area of cross-correlation (p<0.05, see Materials and methods). (**C** and **D**) The black dashed inset (bottom left quadrant in **B**) is then performed in high (**C**) and low (**D**) time constant OFC neurons separately. The sustained coding is present specifically in high time constant cells (largest cluster of cross correlation, p=0.0002), but absent in low time constant cells (p=0.2248; permutation test, see Materials and methods). See also *Figure 5—figure supplement 1 and 2*: Sustained chosen value correlates are present at choice and outcome within DLPFC and ACC, but sustained coding from choice through outcome is

*Figure 5 continued*

absent. Sustained coding between choice and outcome was much stronger in OFC than in DLPFC or ACC (permutation tests; OFC v DLPFC, p=0.0008, OFC v ACC, p<0.0001, see Materials and methods).

The following figure supplements are available for figure 5:

**Figure supplement 1.** Cross-temporal analysis of DLPFC activity.

**Figure supplement 2.** Cross-temporal analysis of ACC activity.

black dashed boxes, permutation tests (see Materials and methods), largest clusters p<0.0001); such sustained coding of value from choice through outcome was absent within DLPFC (*Figure 5—figure supplement 1A*) and ACC (*Figure 5—figure supplement 2A*) neuronal populations. Within OFC, this sustained population code appeared most prominent in the neurons with a high resting time constant τ (*Figure 5C*), but absent in those with a low τ (*Figure 5D*). Note, however, that this difference should be interpreted cautiously, as a formal comparison of cluster size within the high and low τ populations (using a non-parametric permutation test, see Materials and methods) was not significant (p=0.59). Nonetheless, the sustained population code from choice through outcome was much stronger in OFC (*Figure 5B–C*) than in both the ACC and DLPFC populations (*Figure 5—figure supplement 1* and *2*; permutation tests, OFC v DLPFC, p=0008; OFC v ACC, p<0.0001; see Materials and methods). This demonstrates that OFC neurons with persistent activity at rest encode a 'sustained' representation of chosen value until an expected outcome is experienced, and that this neural signature appears unique to OFC.

## Discussion

We have shown that characterising the temporal receptive field of integration of individual PFC neurons based upon their resting activity has significant predictive power for describing their role in decision-making computations. These include the accumulation of evidence during choice, and the persistence of value encoding until the experience of outcome delivery.

Circuits within the prefrontal cortex are endowed with several features that may support persistent activity. These include complex pyramidal cell morphology, strong reciprocal connections, slow-decaying NMDA-Receptor transmission and augmenting synapses (*Wang, 2001*; *Elston, 2003*; *Wang et al., 2006*, *2008*; *Freeman, 1995*; *Wang et al., 2013*). These factors may account for both the prolonged resting stability within PFC, and the ability of its neurons to support computations that subserve flexible cognition (*Miller et al., 1996*). However, there are different cell-classes within the PFC, with substantial heterogeneity in their morphology, synapses and expression of slow-decaying NMDA-Receptors (*Wong and Wang, 2006*; *Zaitsev et al., 2009*; *Wang et al., 2013*). When randomly sampling neurons within the macaque PFC, the morphology, cell-type, cortical layer and synaptic features are unknown. Recorded neurons are therefore likely sampled from separate sub-networks with differing resting stabilities and distinct roles in cognitive processing (*Wang et al., 2013*). This may explain the heterogeneity we observed in both resting activity and involvement in decision making computations observed across PFC neurons. Recent evidence has shown diversity in functional responses of PFC neurons dependent upon the cell-type and cortical layer in which they were located (*Zhou et al., 2012*; *Pinto and Dan, 2015*).

Most importantly in this study, we demonstrated that neurons with higher resting time constants had strong chosen value correlates at choice. Following on from our previous work (*Hunt et al., 2015*) – where we demonstrated that chosen value correlates can arise indirectly from the dynamics of decision processes – our result implies that neurons with more persistent resting activity are more involved in value-based choice. This provides new experimental evidence to support computational theories which attribute evidence integration to strongly recurrent attractor networks (*Wang, 2002*; *Wong and Wang, 2006*). Neurons located within these reverberant PFC subnetworks would be expected to have both higher time constants and stronger value correlates. It also indicates that such models need refinement if they are to encompass the heterogeneous correlates of decisions varaiables that we and others have observed (*Kennerley et al., 2009*; *Meister et al., 2013*). Our

findings facilitate several testable predictions for research into single-neuron mechanisms of decision making. For perceptual decisions, such as the random dot-motion task, which involve the integration of evidence over time more explicitly than our cost-benefit decision paradigm, we would predict task-related neurons would also have high time constants (*Gold and Shadlen, 2007*).

Cross-temporal pattern analysis (*Stokes et al., 2015*) provides a powerful tool to allow for the interrogation of maintained activity within neuronal populations. In addition to decision-making, computational models of working memory also rely upon stable, persistent activity within richly reverberant networks for the retention of information across delays (*Wang, 1999*). Our data showing that evidence maintenance is indeed fulfilled by neurons with higher time constants concurs with this hypothesis. The ability to maintain a representation of chosen value across delays may explain why OFC is essential for delay-based decision making (*Rudebeck et al., 2006*) and why OFC damage causes decision-making and credit assignment deficits (*Rudebeck et al., 2008*; *Noonan et al., 2010*; *Walton et al., 2010*; *Camille et al., 2011*; *Chau et al., 2015*).

Our data on single neuron time constants have provided new insights into potential credit assignment mechanisms within the orbitofrontal cortex. Several imaging and lesion studies have argued that the OFC is involved in the assignment of credit during learning and decision-making (*Walton et al., 2010*; *Takahashi et al., 2011*; *Chau et al., 2015*; *Akaishi et al., 2016*). Single neuron studies have demonstrated OFC cells encode the reward identity across delays (*Lara et al., 2009*), encode specific outcome features during learning (*Raghuraman and Padoa-Schioppa, 2014*; *Lopatina et al., 2015*), and in some cases the same neurons are involved in both choice and outcome processes (*Kennerley and Wallis, 2009*). Indeed, there is a large body of evidence suggesting OFC signals outcome expectancies (*Rangel and Hare, 2010*; *Schoenbaum et al., 2010*). However, despite ideas that OFC is critical for credit assignment during learning, we are not aware of any study that has demonstrated what a neuronal signature of credit assignment might resemble. Here we show that OFC neurons with high temporal specializations not only encode an integrated chosen value signal during choice, but that the same OFC neurons maintain this representation through to the experience of an outcome. This neural signature - when combined with a representation of the chosen stimulus identity, which is also encoded in OFC (*Raghuraman and Padoa-Schioppa, 2014*; *Lopatina et al., 2015*) - could be a key computation for credit assignment processes.

As well as our findings at the single-neuron level, our results reiterate the value of assigning timescales at the level of a cortical area (*Murray et al., 2014*). We replicated the findings of *Murray et al. (2014)* showing that the anterior cingulate cortex (ACC) had the longest timescale within the PFC regions studied. It is possible the ACC may be supporting extended cognitive processes that our experimental paradigm was not designed to capture. These include the encoding or integrating of reward, planning and/or choice information across multiple trials (*Matsumoto et al., 2007*; *Seo and Lee, 2007*; *Bernacchia et al., 2011*; *Hayden et al., 2011*; *Kennerley et al., 2011*; *Stoll et al., 2016*). Future studies might explore the timescales of other prefrontal regions proposed to have unique roles in storing information across multiple trials, such as frontal polar cortex (*Boorman et al., 2009*; *Donoso et al., 2014*).

We demonstrate that calculating the decay in a neuron's intrinsic resting-state autocorrelation can provide a powerful tool for predicting functional properties during cognitive tasks. Our findings therefore have important implications for how neurophysiological datasets are collected and analysed. One current method of avoiding variability in neuronal responses during cognitive tasks is pre-selection of neurons based upon their response properties; neurons with stable, persistent responses on memory guided saccade tasks are preferentially selected for analysis in decision-making tasks (*Roitman and Shadlen, 2002*; *Huk and Shadlen, 2005*; *Yang and Shadlen, 2007*; *Mante et al., 2013*; *Kira et al., 2015*). This method may lead investigators to record from neurons with longer temporal receptive fields, as evidenced (*Murray et al., 2014*) by the higher population-level time constants within the lateral intraparietal area (LIP) when neurons are screened prior to recording (*Freedman and Assad, 2006*) versus when they are not (*Seo et al., 2009*). A more unbiased characterisation of the heterogeneity of neuronal responses may be obtained by recording from all encountered neurons and categorising them post-hoc, as is more common practice in PFC studies (*Freedman et al., 2001*; *Padoa-Schioppa and Assad, 2006*; *Kim et al., 2008*; *Kennerley et al., 2009*; *Hanks et al., 2015*). In the context of decision-making, this has highlighted several 'non-classical' neuronal response profiles in regions such as LIP (*Meister et al., 2013*).

Indeed, even in spite of pre-screening neurons prior to the task, substantial heterogeneity in task-related responses can nonetheless remain (*Premereur et al., 2011*). A more complete understanding of decision-making computations requires us to understand the roles of all of the neurons in these decision processes.

In summary, we have shown that functional specialisation for temporally extended computations predicts the involvement of PFC neurons in specific aspects of value-guided decision making. We anticipate that this approach may become significant in predicting the role of neurons in many other temporally extended computations dependent upon prefrontal cortex. These might include working memory (*Wang et al., 2013*), strategic (*Seo et al., 2014*) and rule-based reasoning (*Buschman et al., 2012*), and foraging behaviours (*Hayden et al., 2011*).

## Materials and methods

Neurophysiological procedures, task structure and regression analysis of single-neuron responses have been reported previously (*Hosokawa et al., 2013*; *Hunt et al., 2015*). In brief, four male rhesus macaques served as subjects. Recordings were taken from dorsolateral prefrontal cortex (DLPFC), orbitofrontal cortex (OFC) and the anterior cingulate cortex (ACC). The sample size of neurons recorded was therefore predetermined from this pre-existing dataset. The number of neurons and cortical areas recorded from each of the four subjects have been reported previously (*Hosokawa et al., 2013*). The regression model for analysing correlates of chosen value was the same as defined previously (*Hunt et al., 2015*).

### Null hypothesis for coefficient of partial determination (*Figure 1*)

A 'null hypothesis' test for the coefficient of partial determination (CPD) was developed to make interpreting results easier. For each behavioural session, a single regressor of interest (e.g. chosen value - *Figure 1A and B*; chosen benefit - *Figure 1—figure supplement 2A*; chosen cost – *Figure 1—figure supplement 2B*), was shuffled across trials and a 'permuted' CPD calculated. This procedure was repeated 1000 times. For each neuron, at each time point, the permuted CPD was averaged across all of the permutations. The null hypothesis CPD for a cortical area was set at the upper bound of the 95% confidence interval across the population.

### Calculation of autocorrelograms (*Figure 2*)

Single-neuron activity during a 1 s fixation period was used to assign time constants. Single unit responses were time locked to the onset of the fixation period of successfully completed trials to create rasters (lasting 1 s from the onset of fixation). The rasters were divided into 20 separate, successive 50 ms bins. The spike count for each neuron within each bin was computed for each trial. We calculated the across-trial correlation of spike counts between all of the bins using Pearson's correlation coefficient. For each *individual neuron*, this produced an autocorrelation matrix when plotted as a function of trial time (e.g. *Figure 2A* left side), or an exponential decay when plotted as a function of time lag between bins (e.g. *Figure 2A* right side).

Using an exponential decay equation (*Murray et al., 2014*), the decay of the autocorrelation with increasing separation time between bins was fitted to the data using the following equation:

$$R(k\Delta) = A\left[exp\left(-\frac{k\Delta}{\tau}\right) + B\right]$$

(1)

In which $k\Delta$ refers to the time lag between time bins (50 to 950 ms) and $\tau$ is the time constant of the cortical area. Neurons from all areas, particularly ACC, showed evidence of lower correlation values at the shortest time lag (50 ms; *Figure 2—figure supplement 2*). This may reflect refractoriness or negative adaptation (*Murray et al. 2014*). To overcome this, fitting started from the largest reduction in autocorrelation (between two consecutive time bins) onwards.

### Assigning a time constant to single neurons (*Figure 2A–B*, *Figure 2— figure supplement 1*)

For most of the key analyses, individual parameters of the autocorrelation decay function in *Equation 1* were estimated for each neuron. Cells with an autocorrelation function poorly fitted by an

exponential decay were excluded from the analysis (see *Figure 2—figure supplement 1* for examples). Initially neurons failing to meet a set of objective criteria were removed (176/857). These criteria were as follows:

1. Fixation firing rate of greater than 1 Hz
2. Decline in the autocorrelation function in the first 250 ms of time lags
3. No 50 ms time-bin within the fixation period with zero spikes across all recorded trials
4. A and B parameters from *Equation 1* cannot both be positive when the autocorrelation function is fitted.

This was followed by a process of visual inspection by two blinded independent observers, where a further set of neurons were considered to possess autocorrelation functions poorly characterised by an exponential decay (235/857 neurons). The autocorrelation functions of all included / excluded neurons are available as supplementary material.

The remaining 446 neurons were assigned a time constant using expectation maximisation in a hierarchical (random effects) fitting procedure. The decay of their resting autocorrelation was fitted using the same equation as above, with $\log(\tau)$, A and B being estimated as a multinomial Gaussian across the neuronal population. Fitting started after the first reduction in autocorrelation between time bins. Neurons from each PFC area were fitted separately.

## Comparing single neuron time constants across cortical areas

Single neuron time constants were log-transformed and grouped by cortical area (DLPFC; OFC; ACC). The variance of these groups was compared using Bartlett's Test. Single neuron time constants were also grouped by cortical area and compared using a Kruskal-Wallis test.

## Assigning time constant at the population level (*Figure 2—figure supplements 2–4*)

Autocorrelation as a function of trial-time and time lag can also be averaged across a population of neurons, prior to fitting *Equation 1* (see *Figure 2—figure supplements 2- 4*). In addition to the data lost due to incomplete trials, previous investigators have excluded a further proportion of trials due to the drifting resting firing rate of neurons over the course of a session (*Murray et al., 2014*; *Nishida et al., 2014*). As we intended to assign time constants to individual neurons, we decided that estimating autocorrelation from a restricted trial number would not provide the best estimation of spike-count autocorrelation. However, it is possible our method artificially inflated autocorrelation due to drifting firing rates throughout a session. Therefore, as a control analysis, we filtered trials when firing rates drifted, using the same approach as in (*Nishida et al., 2014*). For each neuron, the total spike count during the fixation period of each trial was calculated. A sliding window of these spike counts for 100 trials was subdivided into 4 groups of 25 trials and entered into a Kruskal-Wallis test. By shifting this sliding window from the 1st to the last trial within a session, we obtained the longest sequence of trials in which activity did not differ significantly ($p > 0.005$). This procedure reduced the number of trials used for estimating the autocorrelation function on average by 38.4%. When comparing the population level fits to data using the method reported above, very similar time constants were obtained (compare *Figure 2—figure supplement 2* versus *Figure 2—figure supplement 3*).

## Display matrix of chosen value correlates sorted by time constant (*Figure 3A*)

The coefficient of partial determination (CPD) for chosen value (see *Hunt et al., 2015*) across time for each PFC neuron (n = 446) was stacked into a matrix. The rows of the matrix (i.e. each individual neuron) were sorted by increasing time constant, and then smoothed across neurons with a Gaussian kernel, Full Width at Half Maximum=4.5 neurons (S.D. = 2).

## Significance testing using cluster-based permutation tests (*Figure 1A*, *Figure 1B*, *Figure 3B* and *Figure 4*)

To identify significant clusters of chosen value coding whilst correcting for multiple comparisons across time, cluster based permutation tests were used (*Nichols and Holmes, 2002*).

In *Figure 1A and B*, a two-sample T-test compared the chosen value coefficient of partial determination (CPD) at each time bin between two given cortical areas. In *Figure 3B* and *Figure 4*, a two-sample T-test compared the chosen value CPD at each time bin between the median split of neurons with high and low time constants. The longest window of consecutive bins using an uncorrected (cluster-forming) threshold of p<0.01 within a pre-specified time window was then identified. The pre-specified time windows were as follows:

|  | Time window onset | Time window offset |
|---|---|---|
| *Figure 1A*, *Figure 3B* and *4* | Choice epoch onset | Choice epoch offset (1 s after choice epoch onset) |
| *Figure 1B*, *Figure 4* | Reward onset | 1 s after reward onset |

The size of this cluster was compared to a null distribution constructed using a permutation test. Neurons assigned to either each cortical area (*Figure 1*), or high and low time constant groups (*Figure 3B* and *Figure 4*) were randomly permuted 10,000 times and the cluster analysis was repeated for each permutation. The length of the longest cluster for each permutation was entered into the null distribution. The true cluster size was significant at the p<0.05 or p<0.01 level (corrected) if the true cluster length exceeded the 97.5th percentile or 99.5th percentile of the null distribution, respectively.

## Multiple-linear regression

To further test the relationship of chosen value coefficient of partial determination (CPD) with resting time constant, the log-transformed time constant and log-transformed fixation firing rate, along with additional regressors to control for brain area, were regressed onto the log-transformed chosen value CPD of each neuron at the time of the maximal across-area population CPD (410 ms, see *Figure 3B*).

## Cross-temporal pattern analysis (*Figure 5*)

To assess the maintenance and re-emergence of chosen value correlates throughout a trial, we performed a population cross-temporal pattern analysis (*Kennerley et al., 2011*; *Stokes et al., 2013*). This used the same regression model as before (*Hunt et al., 2015*), except that the regression coefficient (Z-score) for each neuron's chosen value coding was calculated separately for odd and even trials. This 'split-half' method was utilised to prevent the analysis being confounded by noise correlations.

A population vector (V), with each entry being the chosen value correlates for *n* cells, was produced for each time point. The population vectors at all of the different time points were then cross-correlated to produce a matrix of correlation coefficients (*Figure 5A*). Each matrix of correlation coefficients was averaged across the diagonal in order for the data to reflect both odd-to-even and even-to-odd trial projections. This analysis was performed on all of the cells within a cortical area, with the analysis also performed separately following a median split of within-area time constant (*Figure 5B–D*, *Figure 5—figure supplements 1–2*). The consistency of the population code between choice and outcome within OFC cells was of particular interest. Therefore, data from the 1 s choice epoch were correlated against two 1 s periods directly preceding and following reward onset, with the results displayed within the grey and black dashed boxes respectively in *Figure 5B* and *Figure 5—figure supplements 1A*, *2A* for all cells, and separately for high and low times constants in *Figure 5C–D* and *Figure 5—figure supplements 1B-C*, *2B-C*).

To demonstrate sustained population coding of chosen value correlates during choice, a cluster-based permutation test was used (*Nichols and Holmes, 2002*). All correlation coefficients with an uncorrected p<0.01 were highlighted. Any area of interconnecting pixels was defined as a true cluster. The ordering of all of the population vectors was then randomised and the analysis repeated. This permutation occurred 10,000 times and produced a null distribution of cluster sizes. True clusters were significant to the p<0.05 (0.01) level if the area of interconnecting pixels exceeded the 97.5th percentile (99.5th) of those in the null distribution.

This analysis was also repeated for the 1 by 1 s periods of reward onset versus reward onset; choice onset versus 1 s prior to reward onset; choice onset versus 1 s following reward onset; 1s pre-reward onset versus 1s following reward onset.

## Comparing sustained coding between cortical areas and high/low time constant neurons (*Figure 5*)

To compare the sustained coding present from choice through reward delivery between different cortical areas, a permutation test was performed. The black dashed area of the cross-temporal population correlation matrices of *Figure 5B*, *Figure 5—figure supplement 1A* and *Figure 5—figure supplement 2A* were extracted. For each pair of brain areas, the cross-temporal correlation coefficients at each corresponding pixel were compared using Fisher's r-to-Z transformation. All pixels which had correlation coefficients which were significantly different between brain areas (with an uncorrected $p < 0.01$) were highlighted. Any area of interconnecting pixels was defined as a true cluster. The largest area of interconnecting pixels was identified and defined as the 'Largest True Difference Cluster'. The assignment of neurons to brain areas was then shuffled and the analysis repeated 10,000 times to produce a null distribution of Difference Cluster sizes, against which the true cluster size was compared. The test was performed independently for OFC v DLPFC, OFC v ACC and DLPFC v ACC.

A similar test was performed to compare high vs. low time constant neurons (i.e. to compare *Figure 5C* vs. *Figure 5D*); except in the permuted data, neurons were shuffled between high/low groups - as opposed to between different brain areas.

## Data availability

Data (and MATLAB scripts to reproduce the analyses shown in this paper) are available from the Dryad Digital Repository: 10.5061/dryad.5b331 (*Cavanagh et al., 2016*)

## Acknowledgements

LTH was supported by a Sir Henry Wellcome Fellowship from the Wellcome Trust (098830/Z/12/Z). JDW was supported by funding from NIMH R01-MH097990 and NIDA R21-DA035209. SWK was supported by NIMH (F32MH081521) and by a Wellcome Trust New Investigator Award (096689/Z/11/Z). SEC was supported by the Middlesex Hospital Medical School General Charitable Trust.

## Additional information

### Funding

| Funder | Grant reference number | Author |
|---|---|---|
| Middlesex Hospital Medical School General Charitable Trust | Graduate Student Fellowship | Sean E Cavanagh |
| National Institute on Drug Abuse | R21-DA035209 | Joni D Wallis |
| National Institute of Mental Health | R01-MH097990 | Joni D Wallis |
| Wellcome Trust | 096689/Z/11/Z | Steven W Kennerley |
| National Institute of Mental Health | F32MH081521 | Steven W Kennerley |
| Wellcome Trust | 098830/Z/12/Z | Laurence T Hunt |

The funders had no role in study design, data collection and interpretation, or the decision to submit the work for publication.

### Author contributions

SEC, Analysis and interpretation of data, Drafting or revising the article; JDW, Conception and design, Acquisition of data; SWK, Conception and design, Acquisition of data, Analysis and

interpretation of data, Drafting or revising the article; LTH, Conception and design, Analysis and interpretation of data, Drafting or revising the article

**Author ORCIDs**
Sean E Cavanagh, http://orcid.org/0000-0001-9275-2725
Laurence T Hunt, http://orcid.org/0000-0002-8393-8533

**Ethics**
Animal experimentation: Ethical approval was obtained for this study. All procedures were in accord with the National Institute of Health guidelines (Assurance Number A3084-01) and the recommendations of the U.C. Berkeley Animal Care and Use Committee (Protocol Number R283).

## Additional files

### Major datasets

The following dataset was generated:

| Author(s) | Year | Dataset title | Dataset URL | Database, license, and accessibility information |
|---|---|---|---|---|
| Cavanagh S, Wallis J, Kennerley S, Hunt L | 2016 | Data from: Autocorrelation structure at rest predicts value correlates of single neurons during reward-guided choice | http://dx.doi.org/10.5061/dryad.5b331 | Available at Dryad Digital Repository under a CC0 Public Domain Dedication |

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
