## [Decision Letter]

Thank you for submitting your article "Autocorrelation structure at rest predicts value correlates of single neurons during reward-guided choice" for consideration by *eLife*. Your article has been reviewed by two peer reviewers, and the evaluation has been overseen by a Reviewing Editor and Eve Marder as the Senior Editor. The following individual involved in review of your submission has agreed to reveal his identity: Daeyeol Lee (Reviewer #1).

The reviewers have discussed the reviews with one another and the Reviewing Editor has drafted this decision to help you prepare a revised submission.

Summary:

This paper builds on the authors' earlier analysis of the single-neuron data previously collected in three different regions of the primate prefrontal cortex (DLPFC, ACC, and OFC), now focusing on the relationship between the intrinsic timescale (time constant) of spiking activity and their role in value-based decision making. In particular, they found that neurons in the ACC and OFC with longer time constants during fixation (as measured by spike rate autocorrelation) are more likely to encode chosen value signals. Moreover, OFC neurons tended to encode the chosen value signals more strongly during the outcome period compared to the neurons in the DLPFC and ACC. The cross temporal correlation analysis also revealed that the chosen value coding in the OFC during the choice and outcome epochs was consistent. The authors suggested that this might be important for the proposed role of the OFC in resolving temporal credit assignment.

Essential revisions:

Overall, the findings reported in this manuscript are very timely. They are nice additions to the previous work from the authors, and provide important new insights into the neural mechanism of decision making. Also, the manuscript is written clearly and easy to follow. Nevertheless, both reviewers identified some important issues that need to be clarified.

1) It is not clear how the persistent coding of chosen value signals can be used for, or reflect, the resolution of the temporal credit assignment problem. For resolving the temporal credit assignment problem, the brain must recognize one of the previous actions or previously visited states that is related to a particular outcome after a delay. How can the chosen value signals, rather than the signals directly related to previous actions or states, can be used for this purpose? The focus on the temporal credit assignment problem also seems a bit inconsistent with the prediction of the authors from the Wang model about the relationship between the timescale and value coding. In other words, their model of mutual inhibition predicts this relationship, even though it is not clear how that model resolves the temporal credit assignment problem.

2) Similarly, the Discussion focuses mostly on the function of the OFC, but the results from this manuscript and Murray et al. showed that the longest time scale is seen in the ACC. Therefore, it might be helpful to include some discussion about the possible function of the long time scale of ACC activity. One possibility is that ACC might play a more important role in integrating signals across multiple trials, as suggested by Seo and Lee (2007) and Bernacchia et al. (2011).

3) Tests of the association between time constant and chosen-value coding mainly use a median split on the time constant tau. However, it doesn't look like tau values fall into discrete high and low clusters (there's no apparent discontinuity at the median in Figure 3, right-hand side). The rank correlation test mentioned in the third paragraph of the Results seems like a much more natural approach. What's the justification for not using the rank correlation for all the analyses, i.e. the tests of the entire time course, of individual brain regions, and of the outcome phase? As it stands the correlation analysis is confined to a single time point, and the criterion for choosing this time point is vague ("maximal population response"). Does this mean (1) maximum spike rate, (2) maximum population-average CPD, or (3) maximum high-versus-low effect in Figure 3? #1 seems like the most natural reading, but #2 is what I would guess given the context (#3 would be circular).

4) In a few places the paper infers differences because an effect reaches significance in one condition but not another. A direct contrast between the two conditions is generally more appropriate in such cases. This applies to: (1) greater outcome-related value coding in OFC than in DLPFC/ACC (Results, fourth paragraph); (2) greater reactivation coding in high-tau than low-tau neurons (Results, last paragraph); (3) greater reactivation coding in OFC than in DLPFC/ACC (Figure 5—figure supplement 1–Figure 5—figure supplement 2).

5) For the cross-temporal correlation analysis, the authors draw a distinction between "sustained" and "reactivation" coding. But this distinction sometimes gets blurry. The evidence mainly supports reactivation coding, but the conclusion is that OFC is "maintaining a representation of chosen value until an expected outcome is experienced" (Results, last paragraph), which sounds more like sustained coding. Similarly, earlier (Results, fourth paragraph) the paper concludes that OFC codes value through the choice-outcome interval, but only a post-outcome epoch is actually tested.

6) The paper should at least briefly address the distinction between "chosen value" and what one might call "outcome value" – the size of the juice reward. These aren't identical, since chosen value also incorporates an effort/delay requirement. But they may be correlated. Can the authors rule out that OFC is merely encoding the juice magnitude in the outcome phase? That is, is there direct evidence it also encodes the (already completed) effort or delay requirement?

---

## [Author Response]

*Essential revisions:*

*Overall, the findings reported in this manuscript are very timely. They are nice additions to the previous work from the authors, and provide important new insights into the neural mechanism of decision making. Also, the manuscript is written clearly and easy to follow. Nevertheless, both reviewers identified some important issues that need to be clarified.*

*1) It is not clear how the persistent coding of chosen value signals can be used for, or reflect, the resolution of the temporal credit assignment problem. For resolving the temporal credit assignment problem, the brain must recognize one of the previous actions or previously visited states that is related to a particular outcome after a delay. How can the chosen value signals, rather than the signals directly related to previous actions or states, can be used for this purpose? The focus on the temporal credit assignment problem also seems a bit inconsistent with the prediction of the authors from the Wang model about the relationship between the timescale and value coding. In other words, their model of mutual inhibition predicts this relationship, even though it is not clear how that model resolves the temporal credit assignment problem.*

This comment prompted us to consider more carefully the implications of a sustained chosen value signal from choice through outcome. We agree that a chosen value signal could only play an important part in resolving the temporal credit assignment problem in combination with a representation of the chosen stimuli/action to which credit must be assigned.

In our original paper (Huntet al., 2015), we found that a signal for chosen action was present in dorsolateral prefrontal cortex (DLPFC) in the latter part of the choice epoch. We therefore tested whether chosen action coding was also present at the time of reward delivery. Just as is evident during the choice epoch, chosen action signals during the outcome epoch are predominantly found in DLPFC:

Author response image 1.Population averages when chosen action was regressed onto firing rate during reward delivery.DLPFC showed stronger chosen action correlates following reward onset than ACC and OFC (permutation tests; DLPFC v OFC, p = 0.0006, DLPFC v ACC, p = 0.0002; see Methods). Dashed lines mark the null hypothesis level for CPD in each cortical area (see Methods).**DOI:**
http://dx.doi.org/10.7554/eLife.18937.017

We also tested whether the presence of this chosen action signal, which might contribute to the resolution of the temporal credit assignment problem, was predominant within a subset of neurons with a particular resting time constant. We found that the strongest coding of chosen action during outcome was found within the high time constant neurons within DLPFC:

Author response image 2.Dorsolateral prefrontal cortex neurons with higher resting time constant code chosen action more strongly around reward onset.(**A**) As in Figure 4 median split of neurons by their resting time constant was performed within DLPFC. The coefficient of partial determination (CPD) for chosen action in high time constant (blue) and low time constant (red) neurons is plotted timelocked to reward onset. CPD (mean ± SE) for chosen action was calculated by multiple linear regression analysis (see Methods). (**B**) As in Figure 4—figure supplement 1, a rank correlation between resting time constant and chosen action coding is plotted. There was a positive correlation between resting time constant and the coefficient of partial determination (CPD) for chosen action at the time of the maximum population-average CPD during outcome (vertical purple line and asterisk, correlation coefficient = 0.1607, p = 0.0249).**DOI:**
http://dx.doi.org/10.7554/eLife.18937.018

This provides some evidence that it may indeed be high time-constant cells that carry the most information about signals relevant for credit assignment at outcome: chosen value (in OFC), and chosen action (in DLPFC).

However, it is important to note that this task requires choices between different stimuli based on their values, rather than action values per se. Although there may have been some credit assignment of outcomes to actions, optimal credit assignment in this task would involve the assignment of values to the visual stimuli. It is possible that firing rates of OFC neurons could encode information about individual stimuli using a linear relationship that also reflected a value code. In our study, such a ‘stimulus identity’ code would be inherently confounded with value, so we do not believe it could be isolated from this dataset. Nevertheless, recent studies (Lopatinaet al., 2015) have demonstrated that single OFC neurons can encode specific stimuli or stimulus-outcome relationships, so it appears plausible that both a stimulus-specific code and a sustained chosen value code could be utilized across the OFC population to support both choice and credit assignment processes.

Nonetheless, we accept the reviewer’s point and have therefore toned down the references to credit assignment throughout the manuscript (except when referencing other papers that address the question of credit assignment more directly). Instead, we refer to this signal as an important component of a system to update stored values.

For example, in the Abstract, we now say:

“Within orbitofrontal cortex, these neurons also sustain coding of chosen value from choice through the delivery of reward, providing a potential neural mechanism for maintaining predictions and updating stored values during learning.”

In the Introduction, we now say:

“This could be one component of a mechanism for credit assignment in learning, which is known to rely upon PFC and in particular orbitofrontal cortex (Walton et al., 2010; Takahashi et al., 2011; Chau et al., 2015; Jocham et al., 2016), with the other component being a representation of the chosen stimulus identity, which is also encoded by OFC neurons (Lopatina et al., 2015).”

In the Results, we now say:

“This implies a unique neuronal signature within OFC which could contribute to the linking of choices to outcomes, a process critical for learning.”

In the Discussion we now add:

“This neural signature – when combined with a representation of the chosen stimulus identity, which is also encoded in OFC (Lopatina et al., 2015) – could be a key computation for credit assignment processes.”

*2) Similarly, the Discussion focuses mostly on the function of the OFC, but the results from this manuscript and Murray et al. showed that the longest time scale is seen in the ACC. Therefore, it might be helpful to include some discussion about the possible function of the long time scale of ACC activity. One possibility is that ACC might play a more important role in integrating signals across multiple trials, as suggested by Seo and Lee (2007) and Bernacchia et al. (2011).*

Thank you for raising this comment. We agree that this is a point worth discussing. We have added the following paragraph to the Discussion:

“As well as our findings at the single-neuron level, our results reiterate the value of assigning timescales at the level of a cortical area (Murray et al., 2014). […] Future studies might explore the timescales of other prefrontal regions proposed to have unique roles in storing information across multiple trials, such as frontal polar cortex (Boorman et al., 2009; Donoso et al., 2014).”

*3) Tests of the association between time constant and chosen-value coding mainly use a median split on the time constant tau. However, it doesn't look like tau values fall into discrete high and low clusters (there's no apparent discontinuity at the median in Figure 3, right-hand side). The rank correlation test mentioned in the third paragraph of the Results seems like a much more natural approach. What's the justification for not using the rank correlation for all the analyses, i.e. the tests of the entire time course, of individual brain regions, and of the outcome phase? As it stands the correlation analysis is confined to a single time point, and the criterion for choosing this time point is vague ("maximal population response"). Does this mean (1) maximum spike rate, (2) maximum population-average CPD, or (3) maximum high-versus-low effect in Figure 3? #1 seems like the most natural reading, but #2 is what I would guess given the context (#3 would be circular).*

We thank the reviewers for raising this important point. As mentioned by the reviewers below, we felt that the median split was the most straightforward way to visualise the difference between chosen value correlates in both low and high time constant cells. We also tried to show a representation of chosen value correlates across the entire population in Figure 3, sorted by time constant. We felt that it was important to show both of these, as they demonstrate that there are some cells with comparatively low time constants that do have some variance explained by chosen value – but they are fewer in number and weaker than those with high time constants.

In response to this comment, however, we have added sliding rank correlation tests for both choice and outcome periods for both Figure 3 and Figure 4 as supplementary figures. In short, these supplementary analyses essentially recapitulate the results shown using the median split approach.

In new Figure 3—figure supplement 1 (collapsed across regions), there is a positive correlation between time constant and chosen value coefficient of partial determination (CPD) around the maximum-population average CPD. This time course corresponds to the greatest separation between the high and low time constant median split in Figure 3.

With respect to the reviewer’s final comment: on each plot we have now marked the maximum population-average CPD with a vertical blue line (i.e. the reviewer is correct that we meant #2). We recognise that this was previously vague within the text. We have clarified we meant #2 within the Results section:

“We further demonstrated this relationship by performing a rank correlation between each neuron’s coefficient of partial determination (CPD) at the time of the maximum population-average CPD with its time constant (Correlation Coefficient = 0.148, p = 0.0018; 95% CI [0.0556, 0.2373], Figure 3—figure supplement 1).”

Note that the ‘dip’ in correlation between chosen value correlates and time constants (at the very start of the trial) is also observed in the median split analysis (Figure 3). This result may seem surprising – it appears very early for such a factor to explain variance in neural firing. However, we interpret this finding in a similar way to the ‘prescient’ pre-stimulus activity observed in (Padoa-Schioppa, 2013). In particular, the firing rate of neurons before trial onset may affect the success of the network in making decisions, and so induce correlations with chosen value pre-stimulus. This is indeed apparent in the analyses of Figure 1 and Figure 1—figure supplement 2, where there is chosen value coding that is slightly higher than chance, pre-stimulus, across all three regions. The negative correlation with time constants suggests that this is in a different set of cells to the high time constant ‘temporal integrators’ that express chosen value most strongly during choice. However, as this point is rather orthogonal to the main point of the paper (and there is far less variance explained by chosen value at this time point anyway), we do not focus on this explicitly in the main text.

We also perform the sliding rank correlation separated by regions, in new Figure 4—figure supplement 1. Here there is a positive correlation between time constant and chosen value coefficient of partial determination (CPD) around the maximum-population average CPD within OFC during choice. The positive correlation between time constant and CPD emerges later within ACC, similar to the median split method in Figure 4. At the time of outcome, there is only a relationship between time constant and chosen value coding within OFC; this is in the form of a strong positive correlation both at the maximum-population CPD around 900ms into the outcome period, and in the period shortly after reward onset.

We are in agreement with the view expressed by the reviewers below, however, that the median split is overall the clearer way to visualise the results. This is especially true for the cross-temporal pattern analysis in Figure 5, which relies upon correlating the regression coefficients of distinct groups of neurons. We therefore feel that it is best to focus predominantly on this median-split method in the main manuscript, and include the sliding rank correlation figures shown above as supplementary analyses.

*4) In a few places the paper infers differences because an effect reaches significance in one condition but not another. A direct contrast between the two conditions is generally more appropriate in such cases. This applies to: (1) greater outcome-related value coding in OFC than in DLPFC/ACC (Results, fourth paragraph); (2) greater reactivation coding in high-tau than low-tau neurons (Results, last paragraph); (3) greater reactivation coding in OFC than in DLPFC/ACC (Figure 5—figure supplement 1–Figure 5—figure supplement 2).*

This is a useful comment which has helped to make our conclusions more thorough. To address this comment, we used non-parametric permutation tests to perform the direct contrasts between the conditions. We did this for all of the three suggestions raised by the reviewer.

For (1), we have added a permutation test to compare chosen value coding across brain regions to Figure 1. Details of this test have been added to the “Significance Testing using Cluster-Based Permutation Tests” subheading on the Methods section:

*“*Significance Testing using Cluster-Based Permutation Tests” (Figure 1, Figure 1, Figure 3 and Figure 4).

To identify significant clusters of chosen value coding whilst correcting for multiple comparisons across time, cluster based permutation tests were used (Nichols & Holmes, 2002).

[…] The length of the longest cluster for each permutation was entered into the null distribution. The true cluster size was significant at the p<0.05 or p<0.01 level (corrected) if the true cluster length exceeded the 97.5^th^ percentile or 99.5^th^ percentile of the null distribution, respectively.”

This test highlighted that chosen value coding following reward onset was significantly stronger in OFC than DLPFC or ACC, which we now describe in the Figure 1 legend:

“OFC showed stronger chosen value correlates following reward onset than ACC and DLPFC (permutation tests; OFC v DLPFC, p = 0.0010, OFC v ACC, p = 0.0028; see Methods).”

For (2), we developed a similar permutation test to compare sustained coding between high and low time constant neurons. Details of this test have been included within the new section of the methods entitled “Comparing sustained coding between cortical areas and high/low time constant neurons (Figure 5)”.

“Comparing sustained coding between cortical areas and high/low time constant neurons (Figure 5)

To compare the sustained coding present from choice through reward delivery between different cortical areas, a permutation test was performed. […] A similar test was performed to compare high vs. low time constant neurons (i.e. to compare Figure 5 vs. Figure 5); except in the permuted data, neurons were shuffled between high/low groups – as opposed to between different brain areas.”

Note that this direct comparison did not produce a statistically significant difference. We now report this in the text:

“Within OFC, this sustained population code appeared most prominent in the neurons with a high resting time constant τ (Figure 5), but absent in those with a low τ (Figure 5). Note, however, that this difference should be interpreted cautiously, as a formal comparison of cluster size within the high and low τ populations (using a non-parametric permutation test, see Methods) was not significant (p=0.59).”

For (3), we developed a similar permutation test to compare reactivation coding across cortical areas. Details of this test were also included within the new section of the methods entitled *“*Comparing sustained coding between cortical areas and high/low time constant neurons (Figure 5)”.This test highlighted sustained coding was stronger in OFC than in DLPFC or ACC, which we now report in the Results section:

“Nonetheless, the sustained population code from choice through outcome was much stronger in OFC (Figure 5) than in both the ACC and DLPFC populations (Figure 5—figure supplement 1 and Figure 5—figure supplement 2; permutation tests, OFC v DLPFC, p = 0008; OFC v ACC, p < 0.0001; see Methods).”

*5) For the cross-temporal correlation analysis, the authors draw a distinction between "sustained" and "reactivation" coding. But this distinction sometimes gets blurry. The evidence mainly supports reactivation coding, but the conclusion is that OFC is "maintaining a representation of chosen value until an expected outcome is experienced" (Results, last paragraph), which sounds more like sustained coding. Similarly, earlier (Results, fourth paragraph) the paper concludes that OFC codes value through the choice-outcome interval, but only a post-outcome epoch is actually tested.*

On reflection, we agree with the reviewers that our use of the terms “reactivation” and “sustained” coding became confusing during the Results section. To clarify whether orbitofrontal cortex activity is better characterised as ‘sustained’ coding from choice through reward delivery, or alternatively a ‘reactivation’ code which re-emerges only in response to reward delivery, we extended our cross-temporal pattern analysis backwards in time to begin 1500ms prior to reward onset. We also extended our permutation tests to incorporate a ‘pre-outcome’ epoch (-1000ms to 0ms after reward onset), in addition to our existing ‘outcome’ epoch (0 to 1000ms after reward onset). We chose these epoch lengths to ensure the analysis was not contaminated by choice activity given the differing lengths of the cost (effort/delay requirement) epochs on different trials.

We found a strong cross-correlation within OFC activity that was present for the whole of our extended analysis epoch. There were large significant clusters of activity within the “Pre-Outcome” epoch (grey dashed area) which extended all the way until around 1500ms after reward onset (which is the end of the reward delivery period) in the “Outcome” epoch (black dashed area). We show these new results in the updated Figure 5.

Such sustained coding from pre- to post-outcome was absent in a similar analysis within the ACC and DLPFC populations (Figure 5—figure supplement 1 and Figure 5—figure supplement 2). These results suggest that OFC exhibits “sustained” coding from choice through outcome, rather than “reactivation” coding, and that this sustained coding is unique to OFC. The Results section has been updated accordingly:

“Crucially, however, there was also evidence for sustained coding: the same neuronal population in OFC at choice encoded chosen value from at least 1000ms before outcome through to 1000ms after outcome (warm colours in Figure 5, grey and black dashed boxes, permutation tests (see Methods), largest clusters p < 0.0001); such sustained coding of value from choice through outcome was absent within DLPFC (Figure 5—figure supplement 1) and ACC (Figure 5—figure supplement 2) neuronal populations. […] This demonstrates that OFC neurons with persistent activity at rest encode a “sustained” representation of chosen value until an expected outcome is experienced, and that this neural signature appears unique to OFC.”

*6) The paper should at least briefly address the distinction between "chosen value" and what one might call "outcome value" – the size of the juice reward. These aren't identical, since chosen value also incorporates an effort/delay requirement. But they may be correlated. Can the authors rule out that OFC is merely encoding the juice magnitude in the outcome phase? That is, is there direct evidence it also encodes the (already completed) effort or delay requirement?*

Thanks for raising this point; this crucial distinction was not addressed in the manuscript. We have now included evidence to show that OFC is encoding the chosen cost (i.e. the effort / delay requirement already completed) at the time of outcome – see new Figure 1—figure supplement 2. We used the same regression model as before, but split chosen value into a benefit and cost component.

“However, this was not the case at the time of outcome, where chosen value correlates predominated in OFC (Figure 1). This value signal at outcome contained information about both the chosen benefit and chosen cost (Figure 1—figure supplement 2).”